# Validity and Prognostic Value of a Polygenic Risk Score for Parkinson’s Disease

**DOI:** 10.3390/genes12121859

**Published:** 2021-11-23

**Authors:** Sebastian Koch, Björn-Hergen Laabs, Meike Kasten, Eva-Juliane Vollstedt, Jos Becktepe, Norbert Brüggemann, Andre Franke, Ulrike M. Krämer, Gregor Kuhlenbäumer, Wolfgang Lieb, Brit Mollenhauer, Miriam Neis, Claudia Trenkwalder, Eva Schäffer, Tatiana Usnich, Michael Wittig, Christine Klein, Inke R. König, Katja Lohmann, Michael Krawczak, Amke Caliebe

**Affiliations:** 1Institute of Medical Informatics and Statistics, Kiel University, University Medical Center Schleswig-Holstein, Campus Kiel, 24105 Kiel, Germany; koch@medinfo.uni-kiel.de (S.K.); krawczak@medinfo.uni-kiel.de (M.K.); 2Institute of Medical Biometry and Statistics, University of Luebeck, University Medical Center Schleswig-Holstein, Campus Luebeck, 23562 Luebeck, Germany; b.laabs@uni-luebeck.de (B.-H.L.); inke.koenig@uni-luebeck.de (I.R.K.); 3Department of Psychiatry, University of Luebeck, 23538 Luebeck, Germany; meike.kasten@neuro.uni-luebeck.de; 4Institute of Neurogenetics, University of Luebeck, University Medical Center Schleswig-Holstein, Campus Luebeck, 23538 Luebeck, Germany; jule.vollstedt@neuro.uni-luebeck.de (E.-J.V.); norbert.brueggemann@neuro.uni-luebeck.de (N.B.); tatiana.usnich@neuro.uni-luebeck.de (T.U.); christine.klein@neuro.uni-luebeck.de (C.K.); katja.lohmann@neuro.uni-luebeck.de (K.L.); 5Department of Neurology, Kiel University, 24105 Kiel, Germany; jossteffen.becktepe@uksh.de (J.B.); g.kuhlenbaeumer@neurologie.uni-kiel.de (G.K.); eva.schaeffer@uksh.de (E.S.); 6Department of Neurology, University of Luebeck, 23562 Luebeck, Germany; ulrike.kraemer@neuro.uni-luebeck.de (U.M.K.); mi.neis@uni-luebeck.de (M.N.); 7Institute of Clinical Molecular Biology, Kiel University, 24105 Kiel, Germany; a.franke@mucosa.de (A.F.); m.wittig@mucosa.de (M.W.); 8Institute of Epidemiology and PopGen Biobank, Kiel University, University Medical Center Schleswig-Holstein, Campus Kiel, 24105 Kiel, Germany; wolfgang.lieb@epi.uni-kiel.de; 9Department of Neurology, University Medical Center Goettingen, 37075 Goettingen, Germany; brit.mollenhauer@med.uni-goettingen.de; 10Paracelsus-Elena-Klinik, 34128 Kassel, Germany; ctrenkwalder@gmx.de; 11Department of Midwifery Science, University of Luebeck, 23562 Luebeck, Germany; 12Department of Neurosurgery, University Medical Center Goettingen, 37075 Goettingen, Germany

**Keywords:** Parkinson’s disease, polygenic risk score, replication, validation, prognostic value, genetic risk

## Abstract

Idiopathic Parkinson’s disease (PD) is a complex multifactorial disorder caused by the interplay of both genetic and non-genetic risk factors. Polygenic risk scores (PRSs) are one way to aggregate the effects of a large number of genetic variants upon the risk for a disease like PD in a single quantity. However, reassessment of the performance of a given PRS in independent data sets is a precondition for establishing the PRS as a valid tool to this end. We studied a previously proposed PRS for PD in a separate genetic data set, comprising 1914 PD cases and 4464 controls, and were able to replicate its ability to differentiate between cases and controls. We also assessed theoretically the prognostic value of the PD-PRS, i.e., its ability to predict the development of PD in later life for healthy individuals. As it turned out, the PD-PRS alone can be expected to perform poorly in this regard. Therefore, we conclude that the PD-PRS could serve as an important research tool, but that meaningful PRS-based prognosis of PD at an individual level is not feasible.

## 1. Introduction

Parkinson’s disease (PD) is the second most common neurodegenerative disorder after Alzheimer’s disease, with a particularly high prevalence seen in Europe and North America [1]. PD has a complex multifactorial etiology in which both environmental and genetic factors play a prominent role. The main risk factor for PD hitherto identified, however, is age, and both prevalence and incidence increase exponentially in later life.

While some 3–5% of PD cases are monogenic, recent genome-wide association studies (GWAS) revealed that idiopathic PD is highly polygenic [2,3,4]. Therefore, the development of polygenic risk scores (PRSs) as a means to summarize the effect of the genetic background upon an individual’s disease risk in a single number appears meaningful for idiopathic PD. Several PRSs have been developed for PD affection status, age-at-onset and specific symptoms in studies of variable size and using different methodologies [2,5,6,7,8,9,10].

Although the construction of a PRS is rather straightforward using existing software, the validation of existing PRSs through an assessment of their performance in independent data sets has still been undertaken only rarely and, to our knowledge, not for PD. One aim of our study therefore was to investigate in more detail the discriminatory power of a PRS for PD previously published by Nalls et al. [2]. This PRS was developed based upon the largest meta-GWAS for the disease to date and comprises 1805 single nucleotide polymorphisms (SNPs). Our second aim was to assess the prognostic value of this PD-PRS. In fact, while PRSs usually differentiate well between cases and controls, their utility for disease prognostics has been a matter of intensive debate [11,12].

## 2. Materials and Methods

### 2.1. Samples

The samples analyzed in the present study originated from five German cohorts comprising a total of 1914 PD cases and 4464 controls after quality control (Table A1). The data sets were collated within the framework of DFG Research Unit ’ProtectMove’ (FOR2488). The samples of two PD patient and control cohorts (Kiel PD, Luebeck PD) were recruited locally in Schleswig-Holstein, the northernmost federal state of Germany. EPIPARK is an additional prospective and longitudinal observational single-center study from Luebeck, focused upon the non-motor symptoms of PD patients [13]. DeNoPa is a prospective and longitudinal observational single-center study from Kassel in central Germany, aimed specifically at improving early diagnosis and prognosis of PD. Participants include early untreated PD patients and matched healthy controls [14]. The PopGen biobank [15,16] is a central research infrastructure, maintained by Kiel University, for the recruitment of case-control cohorts for defined diseases [15,16]. For the present study, PopGen contributed 661 PD patients and 3093 unaffected individuals from the broader Kiel area.

### 2.2. Genotyping, Genotype Imputation and Quality Control

Genomic DNA was extracted from peripheral blood leukocytes and genotyped using the Infinium Global Screening Array with Custom Content (GSA; Illumina Inc., San Diego, CA, USA) which targets 645,896 variants. Quality control was performed with PLINK 1.9, PLINK 2.0 and R package *plinkQC* [17,18,19,20,21,22]. 

At the SNP level, quality control was carried out with thresholds of 0.01 for the minor allele frequency (MAF), of 0.98 for the SNP call rate and of 10^−50^ for the software-issued p value of the Hardy–Weinberg equilibrium test. Some 431,738 variants passed quality control and were used for imputation with *SHAPEIT2* [23] and *IMPUTE2* [24], based upon the public part of the HRC reference panel (release 1.1, The European Genome-Phenome Archive, EGAS00001001710) [25]. Imputation yielded genotype data for a total of 39,106,911 variants and after the exclusion of variants with MAF < 0.01 or an info score < 0.7, some 7,804,284 variants remained for further analyses.

At the participant level, 6794 individuals were initially available from the five cohorts. Individuals with a call rate < 0.98 or with a heterozygosity value > 3 standard deviations different from the mean on the non-imputed data were removed. To exclude potential relatives and population outliers, linkage disequilibrium pruning was performed using a window size of 50 variants, shifted by five variants, and an r^2^ threshold of 0.2, leaving 186,064 variants. Pairwise identity-by-descent (IBD) was then estimated and individuals were removed in a customized selection process (see Section A.1) until all pairwise IBD values were <0.1. For details on the identification of population outliers, see Section A.2 and Figure A1. In total, 416 individuals were removed leaving 6378 individuals (1914 cases, 4464 controls) for further analysis. Principal component analysis (PCA) plots of the samples from our study and from the 1000Genomes project can be found in Figure A2.

### 2.3. Analysis of Parkinson’s Disease Polygenic Risk Score (PD-PRS)

We evaluated a PRS for PD published by Nalls et al. [2]. The list of the 1805 SNPs included in this PD-PRS, together with reference alleles and effect sizes, was kindly provided to us by the first author. Matching the SNPs to our imputed SNPs was done by reference to their chromosomal positions. Some 1743 of the PD-PRS SNPs were represented in our data set, and all of these SNPs were imputed (the 62 omitted SNPs are listed in Table A2). 

The PD-PRS values were standardized by subtraction of the mean and division by the standard deviation of the PD-PRS among controls. This standardized version of the PRS will henceforth be used and also referred to as ‘PD-PRS’ as well. Density plots were created with base-R function *density*. Logistic regression analysis was performed treating the case-control status as outcome and the PD-PRS value as influence variable, adjusted for the first three PCs, sex and age-at-sampling. An additional logistic regression analysis, excluding age-at-sampling, was performed among cases from the lowest and highest age-at-onset quartiles, treating quartile affiliation as outcome. A two-sided significance level of 0.05 was adopted for the Wald test embedded into the logistic regression analysis.

Receiver operating characteristic (ROC) curves and corresponding areas under curve (AUCs) were calculated with R package *pROC* [26] and 95% confidence intervals for odds ratios were constructed with the *oddsratio.wald* function from the *epitools* package [27].

### 2.4. Identification of Most Relevant PD-PRS SNPs

We evaluated which SNPs of the PD-PRS were most relevant for distinguishing cases from controls by determining their influence upon the AUC. This was done in three steps.

The PD-PRS was repeatedly calculated, excluding one SNP each time, and determining the AUC of the PD-PRS without the SNP. These AUCs will be referred to as ‘AUC-SNP’ values.SNPs were sequentially removed from the PD-PRS based upon the steepest decline of the AUC of the remaining SNPs, until the 95% confidence interval of the residual AUC included 0.5. This set of removed SNPs will be referred to as ‘most relevant SNPs’.The results from step 1 and step 2 were combined in a single plot, relating the AUC-SNP values of SNPs (y axis) to their AUC-SNP-based rank (x axis) and color-coding the set of most relevant SNPs from step 2 together with the set of 47 genome-wide significant SNPs identified by Nalls et al. [2] and included in our PD-PRS.

R package *biomaRt* and the hsapiens_gene_ensembl data set from Ensembl were used to identify genes that included at least one of the most relevant SNPs [28,29,30]. Coding and functional information on individual SNPs were obtained from dbSNP [31].

### 2.5. Prognostic Value of PD-PRS

The *coords* function from R package *pROC* [26] was used to derive appropriate PD-PRS thresholds from ROC curves, and to determine the corresponding values of sensitivity and specificity. Thresholds were calculated by maximizing a weighted Youden-Index:max(costs ∙ sensitivity + specificity)
where ‘costs’ was defined as the relative severity of a false negative compared to a false positive result (i.e., classification or prediction as PD). Costs were varied from 1 to 5 in steps of 0.0001.

For fixed specificity and sensitivity, the positive and negative predictive values (ppv, npv) were computed with Bayes formula as
$$ppv=\frac{sensitivity\cdot prevalence}{sensitivity\cdot prevalence+\left(1-specificity\right)\cdot \left(1-prevalence\right)}$$
$$npv=\frac{specificity\cdot \left(1-prevalence\right)}{specificity\cdot \left(1-prevalence\right)+\left(1-sensitivity\right)\cdot prevalence}$$

To evaluate the prognostic value of the PD-PRS, we had to include the residual lifetime incidence in the above formulae instead of the disease prevalence. To this end, we adopted the age-specific incidence and death rates *I*_[interval]_ and *D*_[interval]_ from the SIa strategy in [32]. The SIa strategy used only cases with at least two diagnoses of PD to avoid false positive diagnoses. *I*_[interval]_ and *D*_[interval]_ were given for 5-year age intervals, starting from [50–54] and ending with [95+]. Since the death rates were given as annual probabilities to die within a given interval, the probability to survive that interval can be approximated by *S*_[interval]_ = (1 − *D*_[interval]_)^5^. For individuals from a given age interval [d,d+5], the residual lifetime incidence can then be computed as
*I*_[d, 95+]_ = *I*_[d, d+5]_ + (*I*_[d+6, d+11]_*∙S*_[d, d+5]_∙(1 − *I*_[d, d+5]_)) + … *+* (*I*_[95+]_*∙S*_[d, d+5]_∙…∙*S*_[90, 94]_∙(1 − *I*_[d, d+5]_)∙ … ∙(1-*I*_[90, 94]_)).

The resulting residual lifetime incidence values are listed in Table A3.

## 3. Results

### 3.1. Validation of Published Parkinson’s Disease Polygenic Risk Score (PD-PRS)

To independently validate the (standardized) PD-PRS proposed by Nalls et al. [2], we investigated the performance of this PRS in a separate data set comprising 1914 PD cases and 4464 controls (Table A1). The distribution of the PD-PRS clearly differed between the two groups (Figure 1A; Wald test *p* < 10^−5^, Table 1). Nagelkerke’s pseudo-R^2^ from the logistic regression analysis equaled 0.35 when including PD-PRS, sex, age and the first three principal components (PCs), and 0.30 when the PD-PRS was not included (Table 1). The area under curve (AUC) for the receiver operating characteristic (ROC) curve (Figure 1B) was 0.65, which was comparable to the AUC obtained in the original study [2]. The disease odds ratios (ORs) for the 2nd to 10th deciles of the PRS distribution among controls ranged from 1.26 (2nd decile) to 6.10 (10th decile; 1st decile used as reference; Figure 2).

The PD-PRS was also able to distinguish well between cases from the 1st and 4th age-at-onset (AAO) quartile (≤54 years vs. >70 years, Figure 3A, *p* = 1.61 × 10^−5^, Table 1). Nagelkerke’s pseudo-R^2^ from the logistic regression was 0.039 including PD-PRS, sex and the first three PCs, and 0.009 when the PD-PRS was not included. The AUC of the ROC equaled 0.59 (Figure 3B, Table 1) and was hence considerably smaller than the AUC obtained for distinguishing cases from controls.

### 3.2. Most Relevant SNPs in PD-PRS

We identified 422 SNPs as being the most relevant for distinguishing cases from controls, judged by their influence upon the AUC in a backward-selection process (see Methods). Of these SNPs, 287 are located within a gene. Table 2 lists the top 20 most relevant SNPs inside genes (for a complete list, see Table A4). Of all 1743 SNPs analyzed, some 47 had been genome-wide significant in the meta-GWAS by Nalls et al. [2]. Thirty-two of these (68%) were among the 422 most relevant SNPs identified here, and 25 of them (78%) were intra-genic. When all 1743 SNPs were ranked according to the AUC obtained when a given SNP was removed (Figure 4), the 422 most relevant SNPs occurred mostly on the left side of the graph meaning that the AUC is strongly reduced upon the removal of the SNP. The 32 most relevant and genome-wide significant SNPs, in particular, were found to cluster at the far left of the graph.

### 3.3. Prognostic Value of PD-PRS

To investigate the prognostic value of the PD-PRS, an individual was defined as ‘test-positive’ if their PRS exceeded a given threshold of the PRS and ‘test-negative’ if not. Thus, sensitivity in this context means the probability that a person who develops PD in later life has a PRS above the threshold while specificity is the probability that a person who will not develop PD during their lifetime is test-negative. Since sensitivity is generally more important than specificity for screening tests, we considered different relative costs of false negative vs false positive test results when maximizing a weighted Youden index to determine the optimal PD-PRS threshold (Table 3). For costs of 1, i.e., when false positives and false negatives are deemed equally serious, the optimal PD-PRS threshold equaled 0.33, yielding a sensitivity of 0.58 and a specificity of 0.63. For costs of 5, the sensitivity equaled 1 and the specificity equaled 0.003 at an optimal PD-PRS threshold of −2.667 (Table 3, Figure 5A).

For fixed costs, the age-specific predictive values of the PD-PRS differed only little up to age interval [70–74], after which the positive predictive value (ppv) declined and the negative predictive value (npv) increased (Table 4, Figure 5B). Across all age groups and costs levels, the ppv was very low with a maximum of 0.027 up to 74 years at costs of 1. The minimum ppv was 0.005 for the highest age group (90+) at costs of 5. The npv varied between 0.988 (≤74 years, costs 1) and 1 (all age groups, costs 5).

## 4. Discussion

In the present study, we replicated the performance of the PD-PRS developed by Nalls et al. [2] in an independent data set. It turned out that the PD-PRS was clearly able to distinguish between cases and controls and that it was increased in cases of early age-at-onset. Individuals in the 10th PRS decile had an OR of around 6 of having PD as compared to individuals in the lowest decile. This is in line with the results by Nalls et al. [2] who reported ORs of 3.74 and 6.25 for the highest quartiles in their two data sets. The most relevant PRS SNPs identified in our study included many genome-wide significant SNPs from the Nalls et al. study [2], as was to be expected. In fact, of the 47 genome-wide significant SNPs, some 32 (68%) were found to be most relevant in the sense of our study. However, this is still only a small fraction (7.5%) of the total number of 422 most relevant SNPs, which highlights the polygenic background of PD with several low-effect variants and justifies the fact that not only genome-wide significant SNPs were originally included in the PRS.

In the recent past, the research community has become increasingly aware of the problem of non-replicability of research findings in independent data sets or with different methods [33]. This has been termed the “replication crisis” or “reproducibility crisis” [34,35]. Studies aiming at validating existing PRSs are still rare and, usually, new data set-specific PRSs are developed instead because this is easy with existing software. Nevertheless, PRS replication should be mandatory [36] and our replication of the results reported by Nalls et al. [2], in an independent data set, is reassuring. It supports the idea that this PD-PRS can be used to capture the contribution of the genetic background of an individual to their PD risk. The PD-PRS could hence be a valid instrument to adjust for the genetic background component in statistical models for PD. Moreover, it may also facilitate studies of the genetic overlap between different diseases or disease subtypes and of the interaction between genetic and environmental factors.

It has to be kept in mind, however, that PRSs only capture the effect of common genetic variants. Highly-penetrant rare or private variants as well as other types of variations such as copy number variants or indels are not represented [37]. Another drawback of PRSs is their dependency on the ancestry of populations [38]. The PD-PRS analyzed in the present study was both constructed and validated in populations of European ancestry, and transferability of the results to other ancestries cannot be taken for granted but has to be investigated in future studies. On a related note, it must be kept in mind that all PD-PRS SNPs considered in our study were imputed. This does not seem to have impaired our replication of the results of Nalls et al. [2], probably due to our stringent quality control. For populations, where a good imputation reference is lacking, consistent PRS performance may not be taken for granted.

Quality control in our study led to the exclusion of 62 of the original 1805 PD-PRS SNPs. The omitted SNPs showed on average a larger effect size in the original meta-analysis than the SNPs included in our PRS (Table A2). The former were excluded mostly (79%) because of very low MAF and the rest because the info score was below 0.70. Despite the higher effect sizes, it is therefore not clear if the additional usage of the 62 SNPs would enhance the performance of the PD-PRS because of low MAF and perhaps difficult imputation. The loss of variants from the score due to difficulties in imputation is a good argument for the adoption of the development of standardized PRSs based on reference variants which are available in common genotyping arrays. This would reduce the imputation problem. 

Whereas PRSs deserve a role in etiological research and statistical modelling of diseases, their prognostic value is dubious [11,12,36]. PRSs are developed to differentiate between cases and controls. Although the level of differentiation achieved is reasonable at a group level, the obtained AUCs are usually insufficient for individual diagnostic or prognostic testing, where an AUC > 0.90 is required [11]. In this study, we evaluated the prognostic value of a specific PD-PRS and calculated its sensitivity and specificity as well as its predictive values for various assumptions about the relative importance of mis-prognoses. Our results were in accordance with the generally held view that a prognostic application of PRSs alone is not meaningful. The negative predictive values were high which means that people with a low PRS can be reasonably sure not to develop PD, at least not of the type considered in this study. However, the positive predictive values were only of the order of a few percent which means that the probability of a person with a high PRS developing the disease is quite low. Here, the comparison to a hypothetical test which gives everybody a negative test result is helpful: Assuming a lifetime incidence of 5% [39], the negative predictive value of this (nonsense) test would be 95%, i.e., quite similar to a test based solely on the PD-PRS.

There are three ways in which a prognostic test for PD, or any other disease, could potentially help to reduce incidence or severity: change of lifestyle factors, enhanced surveillance or preventive treatment. Of these, a change towards a healthier lifestyle is always meaningful, both from an individual and a population health perspective, and only a test with a positive predictive value much higher, for example, than that of the PD-PRS would mean an additional individual incentive for change. Moreover, with a low incidence and positive predictive value, frequent medical screening of individuals with a high PRS would mean spending valuable resources for individuals who have only a probability of a few percent to actually develop the disease in question. The same holds true for possible preventive treatment if such treatment were available in the first place. Apart from economic constraints, side-effects might result in a negative benefit-risk balance when the incidence of the disease in question is as low as for PD.

A limitation of our study has been that the predictive values were only calculated from theoretical models and were not based directly upon empirical observations. This is a general drawback when evaluating the prognostic value of PRSs because adequate long-term studies would be time-consuming, require large sample sizes and would hence be rather expensive. This notwithstanding, PRSs have to be externally validated and compared to other (clinical) risk models in a clinically meaningful prospective set-up [12,36] because this is a *conditio sine qua non* for the applicability in practice of any prognostic marker. Only a few studies have taken first steps in this direction [40,41,42], and most have found none or only little additional prognostic value of PRSs over and above clinical and demographic predictors. To our knowledge, no such study has been performed yet for PD, where the combination of a PRS with established prodromal markers [43] might be specifically worth investigating in future prospective studies.

## 5. Conclusions

The PD-PRS proposed by Nalls et al. [2] could be validated independently in German patients and controls, suggesting that the PRS may be a meaningful research tool to investigate and adjust for the polygenic component of PD. Individual risk prediction using the PD-PRS alone is, however, not meaningful.

## Figures and Tables

**Figure 1 genes-12-01859-f001:**
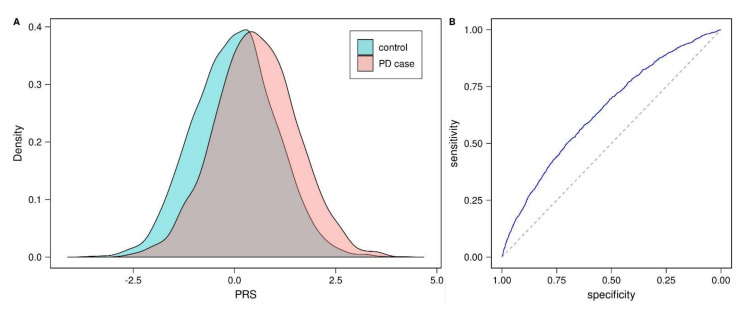
PD-PRS in PD cases and controls. (**A**) Density of PD-PRS in cases and controls. (**B**) ROC curve for PD-PRS as a predictor of case-control status. PRS: polygenic risk score, PD: Parkinson’s disease, ROC: receiver operating characteristic.

**Figure 2 genes-12-01859-f002:**
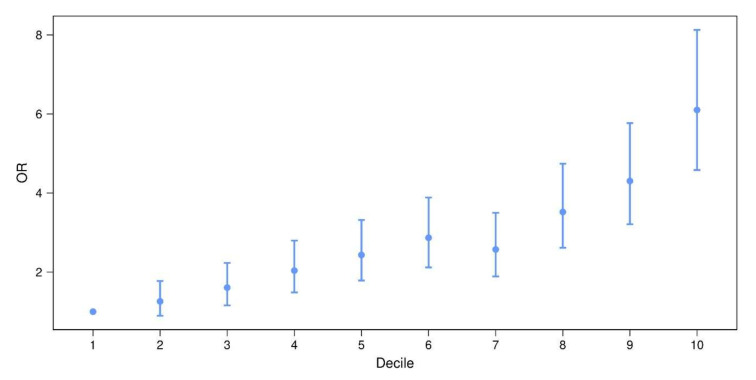
Disease OR for the 2nd to 10th deciles of the PD-PRS distribution among controls. (1st decile used as reference). Vertical bars demarcate 95% confidence intervals. OR: odds ratio, PD: Parkinson’s disease, PRS: polygenic risk score.

**Figure 3 genes-12-01859-f003:**
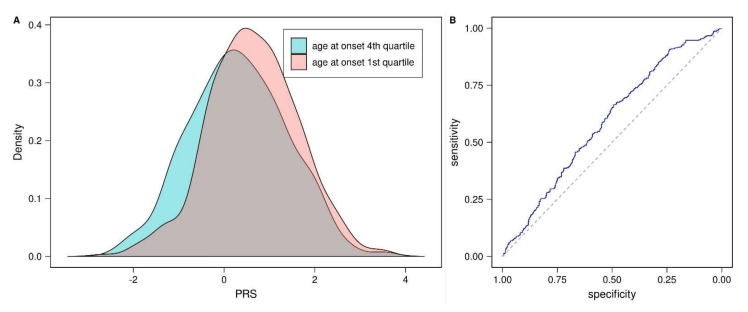
PD-PRS in early and late onset cases. (**A**) Density of PD-PRS in the 1st and 4th AAO quartile of cases. (**B**) ROC curve for PD-PRS as a predictor of 1st vs 4th AAO quartile. AAO: age-at-onset, PRS: polygenic risk score, PD: Parkinson’s disease, ROC: receiver operating characteristic.

**Figure 4 genes-12-01859-f004:**
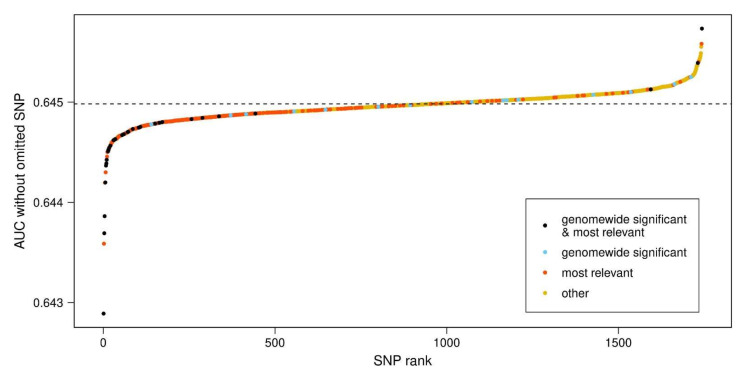
Influence of individual SNPs upon PD-PRS performance. For each of the 1743 PD-PRS SNPs, the AUC was calculated after removing the SNP from the PRS. SNPs were color-coded as either genome-wide significant in a meta-GWAS [2] (blue), as ‘most relevant’ in the present study (red), both of the former (black) or none of the former (yellow). SNP: single nucleotide polymorphism, PD: Parkinson’s disease, PRS: polygenic risk score, AUC: area under ROC curve, ROC: receiver operating characteristic, GWAS: genome-wide association study.

**Figure 5 genes-12-01859-f005:**
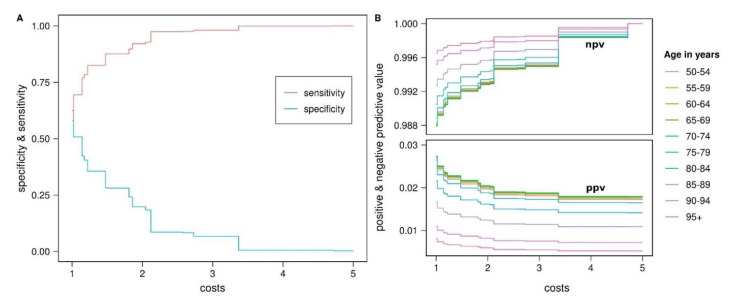
Prognostic value of PD-PRS. (**A**) Sensitivity and specificity of PD-PRS for the optimal threshold were determined by maximizing a weighted Youden index. The relative costs of false negative vs false positive results varied from 1 to 5. (**B**) ppv and npv were calculated from the costs-based sensitivity and specificity and the residual lifetime incidence (see Methods and Table A3) in 10 age groups. PRS: polygenic risk score, PD: Parkinson’s disease, ppv: positive predictive value, npv: negative predictive value.

**Table 1 genes-12-01859-t001:** Comparative validation of PD-PRS.

Data Set	Samples (N)	SNPs (N)	AUC [95% CI]	Nagelkerke’s Pseudo-R^2 a^	*p* Value ^b^	Nagelkerke’s Pseudo-R^2 c^
This study (case/control)	6378	1743	0.645 [0.630, 0.660]	0.348	<10^−5^	0.298
Nalls training ^d^ (case/control)	11,243	1809	0.640 [0.630, 0.650]	n.a.	<10^−5^	n.a.
Nalls validation ^e^ (case/control)	999	1805	0.692 [0.660, 0.725]	n.a.	<10^−5^	n.a.
This study (AAO) ^f^	836	1743	0.590 [0.551, 0.629]	0.039	1.6 × 10^−5^	0.009

^a^ From logistic regression analysis of PD case-control status (first line) and AAO 1st vs 4th quartile (fourth line), each time including PD-PRS, sex, age (only for the analysis of case-control status) and the first three PCs as independent variables. Nalls et al. [2] used a different approach to evaluate logistic regression models, hence a comparison of pseudo-R^2^ is not meaningful. ^b^
*p* value for PD-PRS as an independent variable in the logistic regression analysis (Wald test). ^c^ Same logistic regression model as before, but without PD-PRS as an independent variable. ^d^ NeuroX-dbGaP data set (5851 cases, 5866 controls). ^e^ Harvard Biomarker Study (527 cases, 472 controls). ^f^ Samples belonging to the 1st and 4th AAO quartile among cases analyzed in this study. PD: Parkinson’s disease, PRS: polygenic risk score, SNP: single nucleotide polymorphism, AUC: area under ROC curve, CI: confidence interval, AAO: age-at-onset, ROC: receiver operating characteristic, n.a.: not available.

**Table 2 genes-12-01859-t002:** Top 20 most relevant SNPs located within genes.

HGNC Symbol ^1^	Chr	AUC	Start ^2^	End ^3^	SNP Position ^4^	A1 ^5^	A2 ^6^	GS ^7^	SNP Type
ENSG00000251095	4	0.643	90,472,507	90,647,654	90,626,111	G	A	yes	intron
*SNCA*	4	0.641	90,645,250	90,759,466	90,684,278	A	G	no	intron
*HIP1R*	12	0.640	123,319,000	123,347,507	123,326,598	G	T	yes	intron
*TMEM175*	4	0.639	926,175	952,444	951,947	T	C	yes	missense
*SNCA*	4	0.638	90,645,250	90,759,466	90,757,294	A	C	no	intron
*ASH1L*	1	0.637	155,305,059	155,532,598	155,437,711	G	A	no	intron
*UBQLN4*	1	0.634	156,005,092	156,023,585	156,007,988	G	A	no	intron
ENSG00000225342	12	0.633	40,579,811	40,617,605	40,614,434	C	T	yes	n.a.
*LRRK2*	12	0.633	40,590,546	40,763,087	40,614,434	C	T	yes	n.a.
*STX1B*	16	0.632	31,000,577	31,021,949	31,004,169	T	C	no	synonymous
*INPP5F*	10	0.631	121,485,609	121,588,652	121,536,327	G	A	yes	intron
*CCSER1*	4	0.631	91,048,686	92,523,064	91,164,040	C	T	no	intron
*SLC2A13*	12	0.630	40,148,823	40,499,891	40,388,109	C	T	no	intron
*FBXL19*	16	0.630	30,934,376	30,960,104	30,943,096	A	G	no	intron
ENSG00000251095	4	0.629	90,472,507	90,647,654	90,619,032	C	T	no	intron
*CAB39L*	13	0.629	49,882,786	50,018,262	49,927,732	T	C	yes	intron
*STK39*	2	0.628	168,810,530	169,104,651	168,979,290	C	T	no	intron
*CCT3*	1	0.628	156,278,759	156,337,664	156,300,731	T	C	no	intron
ENSG00000225342	12	0.627	40,579,811	40,617,605	40,614,656	A	G	no	n.a.
*LRRK2*	12	0.627	40,590,546	40,763,087	40,614,656	A	G	no	n.a.

^1^ HGNC symbol or Ensemble gene ID if there is no HGNC symbol available. ^2^ Base pair position of start of gene. ^3^ Base pair position of end of gene. ^4^ Genomic position of SNP. ^5^ Major SNP allele. ^6^ Minor SNP allele. ^7^ Genome-wide significant (GS) in the meta-GWAS by Nalls et al. [2]. HGNC: HUGO Gene Nomenclature Committee, Chr: Chromosome, AUC: area under ROC curve, ROC: receiver operating characteristic, PRS: polygenic risk score, PD: Parkinson’s disease, n.a.: not available.

**Table 3 genes-12-01859-t003:** Prognostic value of PD-PRS.

	Costs				
	1	2	3	4	5
Sensitivity [95% CI]	0.581 [0.479, 0.733]	0.921 [0.880, 0.981]	0.981 [0.973, 1]	0.999 [0.983, 1]	1 [0.996, 1]
Specificity [95% CI]	0.625 [0.472, 0.725]	0.198 [0.075, 0.289]	0.067 [0.004, 0.096]	0.006 [0.002, 0.082]	0.003 [0.002, 0.034]
Threshold ^1^	0.330	−0.868	−1.507	−2.533	−2.667

^1^ Optimal threshold for PD-PRS as determined by maximizing a weighed Youden index. PD: Parkinson’s disease, PRS: polygenic risk score, CI: confidence interval.

**Table 4 genes-12-01859-t004:** Costs- and age-dependent PD-PRS predictive values.

		Costs									
		1		2		3		4		5	
		ppv	npv	ppv	npv	ppv	npv	ppv	npv	ppv	npv
**Age group (Years)**	50–54	0.026	0.988	0.020	0.993	0.018	0.995	0.017	0.998	0.017	1
	55–59	0.027	0.988	0.020	0.993	0.018	0.995	0.018	0.998	0.018	1
	60–64	0.027	0.988	0.020	0.993	0.019	0.995	0.018	0.998	0.018	1
	65–69	0.027	0.988	0.021	0.993	0.019	0.995	0.018	0.998	0.018	1
	70–74	0.027	0.988	0.020	0.993	0.019	0.995	0.018	0.998	0.018	1
	75–79	0.025	0.989	0.019	0.993	0.017	0.995	0.017	0.999	0.016	1
	80–84	0.022	0.990	0.016	0.994	0.015	0.996	0.014	0.999	0.014	1
	85–89	0.017	0.993	0.012	0.996	0.011	0.997	0.011	0.999	0.011	1
	90–94	0.011	0.995	0.008	0.997	0.008	0.998	0.007	0.999	0.007	1
	95+	0.008	0.996	0.006	0.998	0.005	0.999	0.005	1.000	0.005	1

PRS: polygenic risk score, PD: Parkinson’s disease, ppv: positive predictive value, npv: negative predictive value.

## Data Availability

The data that support the results of this study are available upon reasonable request from the corresponding author.

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
