# Peer review of "Validity and Prognostic Value of a Polygenic Risk Score for Parkinson’s Disease"

_genes, 2021, doi:10.3390/genes12121859_

Round 1

Reviewer 1 Report

This paper takes a simple but important task, replicating a PRS result in a new cohort, and conducts the analysis in an exemplary manner (I would use this as an example of how to conduct such an analysis).

I have two minor comments that the authors might wish to consider:

1. I found the use of numbers as indicators of footnotes in Table 1 confusing - it might be clearer to use letters

2. The loss of variants from the score due to difficulties in imputation is a good argument for the adoption of reference-standardised polygenic risk scores (which would reduce this problem) - the authors might want to consider adding a brief discussion of this point to their conclusions.

Reviewer 2 Report

The present work replicates a previous study in an independent sample. Nalls et al. proposed a PD-PRS (polygenic risk score) of 1809 SNPs out of a GWAS with 11243 participants. The reviewed study recruited 6378 participants and after a stringent quality control, was able to determine 1743 SNPS available also in the Nalls study.

The authors are aware of the dependency of thr PRSs on the ancestry of populations and took care of selecting participants of European ancestry as it was also done in Nalls et al study.

The results show that the PD-PRS is able to distinguish between cases and controls and the authors suggest that the PRS may be a meaningful research tool to investigate and adjust for the polygenic component of PD.

The subject of the manuscript is of interest as a continuation of the big GWAS studies. What is more, it highlights the importance of validation in an independent cohort.

The Manuscript is very nicely and friendly written. Methods are explained with detail and supplementary material is as well of most interest.